# Transcriptional Profiling to Assess the Effects of Biological Stimulant Atlanticell Micomix on Tomato Seedlings Under Salt Stress

**DOI:** 10.3390/plants14081198

**Published:** 2025-04-11

**Authors:** María Salud Justamante, Eduardo Larriba, Ernesto Alejandro Zavala-González, Almudena Aranda-Martínez, José Manuel Pérez-Pérez

**Affiliations:** 1Instituto de Bioingeniería, Universidad Miguel Hernández, 03202 Elche, Spain; mjustamante@umh.es (M.S.J.); elarriba@umh.es (E.L.); 2Atlántica Agrícola S.A., 03400 Villena, Spain; ezavala@atlanticaagricola.com (E.A.Z.-G.); aaranda@atlanticaagricola.com (A.A.-M.)

**Keywords:** biostimulants, differentially expressed genes, moderate salinity, *Solanum lycopersicum*, transcriptome profiling

## Abstract

Recent environmental changes in the Mediterranean region, attributable to anthropogenic climate change, present a substantial challenge to the adaptive evaluation of crops and the development of novel improvement strategies. In this study, we established a hydroponic tomato cultivation protocol under in vitro conditions to analyze the transcriptomic profile of seedlings exposed to salinity stress. The study also examined the impact of Atlanticell Micomix, a biological stimulant derived from a mixture of mycorrhizal microorganisms and rhizobacteria, on plant growth and development under standard conditions and in response to moderate salinity. Our transcriptomic analysis indicated a differential effect of biostimulant inoculation compared to the effect induced by salinity stress, involving genes such as *GOX3* or *DIR1*, which are associated with the plant’s defense response to adverse conditions. In addition, the presence of a cross-regulatory module between jasmonic acid and auxin, involving potential orthologs of *IAA29* and *JAZ*, was proposed. The application of the biostimulant demonstrated a potential priming effect on the tomato seedlings, which might be useful in reversing the transcriptomic effects caused by salt stress. A comprehensive analysis of the pathways differentially affected by the treatments facilitates further investigation into the mechanisms underlying these effects.

## 1. Introduction

Currently, the agricultural sector faces the challenge of increasing food production to meet the demands of a growing global population while minimizing its impact on ecosystems and human health, particularly in the context of an accelerated anthropogenic climate change scenario [1,2]. Climate change has a major impact on global precipitation patterns, which is expected to result in longer drought periods in some regions, such as the countries in the Mediterranean basin [3,4]. This region is especially affected by the salinization of agricultural soils [5,6]. The use of fertilizers and pesticides has been essential to increase yield and ensure crop production [7]. However, new methods and approaches have been proposed to enhance the sustainability of agricultural production by reducing synthetic chemical compounds and helping crops to adapt to challenging abiotic conditions [8].

The use of plant biostimulants is a promising and environmentally friendly innovation that enhances the various processes of plant development, including vegetative growth, flowering, and fruit production. Furthermore, plant biostimulants can enhance nutrient uptake efficiency and improve tolerance to biotic and abiotic stress factors, such as those associated with anthropogenic climate change [9,10]. The development of new biostimulants requires precise testing and confirmation of their effects on the morphological and physiological traits of the crop for which they have been designed. Additionally, a detailed understanding of the mechanism of action of the formulations on the host plant is necessary [11,12]. Several authors have employed evaluation approaches for new biostimulant products based on omics technologies [13,14,15,16], which are more efficient and cost-effective. Biostimulants based on microorganisms, mainly fungi and bacteria, are used in agriculture to improve crop quality, production, and stress tolerance. One promising type of biostimulants is formulated with arbuscular mycorrhizal fungi (AMF). These AMF are commonly found in soils and form symbiotic associations with 80% of terrestrial plant species, including most crops [17,18]. These fungi are useful as biofertilizers in organic agriculture due to their ability to improve water and nutrient absorption and increase resistance to abiotic and biotic stresses, particularly in resource-limited environments [19].

In 2023, Spain ranked 9th in tomato worldwide production, with a total volume of 3968 million tons of tomatoes produced on just over 50,090 hectares of land [20]. Recent studies have shown that AMF colonization has positive effects on important agronomic traits in this species, such as fruit quality [21,22] and resistance to aphid infection [23]. It also leads to increased production and better efficiency in responding to biological stress situations. Previous studies have shown that AMF can decrease the accumulation of certain viruses, such as tomato bushy stunt virus and tomato mosaic virus, by upregulating the genes of pathogenesis-related protein in colonized plant tissues [24]. Furthermore, AMF colonization has been observed to increase the endogenous production of compounds derived from alkaloids and fatty acids, resulting in higher mortality rates of herbivorous insects feeding on colonized tomato plants [25,26]. These and other results [27] demonstrate the enhanced disease resistance induced by AMF colonization in tomatoes. The impact of AMF symbiosis on the response to abiotic stresses such as drought, salinity, and high temperatures, has been also studied in this species [28,29,30,31]. Tomato plants colonized with AMF showed the differential regulation of some drought response genes in both the roots and leaves during water deficit conditions [30,32]. High salinity stress can reduce growth and water absorption in many crops, including tomatoes [33,34]. Inoculating tomato seedlings with AMF from species such as *Piriformospora indica* [35,36] or *Rhizophagus irregularis* [37] can enhance salt stress tolerance through various mechanisms. These mechanisms include improved nutrient uptake, increased fatty acid and phospholipid contents, and reduced sodium accumulation by regulating the expression of specific genes [35,36,37]. These findings indicate that inoculation with AMF can enhance plant tolerance and growth under drought and high salinity conditions, as well as activate natural plant defenses. This process is known as priming and is regulated by metabolic and hormonal pathways [38]. To comprehend the effect of AMF inoculation on plant metabolism, it is helpful to analyze changes in the transcriptomic profile of AMF-colonized and non-AMF-colonized plants when exposed to different stressors. Through this approach, several studies have identified the key regulatory genes that control endogenous responses to heavy metals [39], high salinity [40], and low phosphate availability [41]. Recent studies have been conducted to ascertain the impact of plant growth-promoting rhizobacteria (PGPR) with biostimulant properties on the germination and growth of tomato seedlings. These investigations aim to elucidate the role of PGPR in mitigating the deleterious effects of various stresses, including salinity, drought, and elevated temperatures [42,43,44,45,46,47].

Numerous studies have demonstrated the multiple advantages of AMF and/or PGPR inoculation in tomato plants [48,49,50], as well as other crops [51,52]. However, their practical implementation in field conditions as biostimulants requires a thorough comprehension of the mechanisms involved in the plant physiological responses during mycorrhization. Identifying key molecular factors is crucial to fully exploiting the potential of these microorganisms as potential biostimulants. In addition, to counteract the harsh environmental conditions driven by current climate change, it is crucial to conduct a comprehensive assessment of the mechanisms that mediate plant adaptive responses, including enhanced nutrient absorption, improved water efficiency, and resistance to biotic and other abiotic stresses. These mechanisms should be evaluated while considering the characteristics of the cultivars of interest and utilizing the most suitable symbiotic combinations. The application of assays using omics technologies, such as transcriptomics or metabolomics, will provide new insights into the mechanisms that mediate the interaction and response of plants to different biostimulants and under different stress conditions.

In this study, we propose the utilization of an in vitro hydroponic culture system for the implementation of transcriptomic studies in tomato seedlings. The experimental design is validated by studying young tomato seedlings subjected to salt stress and control conditions, in the presence and absence of a commercial biostimulant. A comprehensive analysis of the transcriptomic outcomes will ascertain whether the influence of the biostimulant treatment on salinity stress alleviation is attributable to a direct or indirect effect of the biostimulant on the stress response pathways under investigation.

## 2. Results

### 2.1. Growth Responses of Young Tomato Seedlings to Atlanticell Micomix and Salt Stress

After 2–3 days of transferring tomato seedlings to the hydroponic medium, hyphae were observed in the nutrient solution only in the treatments containing the Atlanticell Micomix solution, hereafter referred to as the AMM solution (Appendix A and see Section 4). To confirm the effective establishment of mycorrhizae, the presence of intracellular arbuscules in the root cortex was analyzed in five roots of each of the four treatments after five days of growth in hydroponic culture. These structures were observed only in treatments containing AMM in all cases (Figure 1a). No significant differences were observed between the treatment with AMM and the treatment with AMM and 100 mM NaCl.

Experiments E1 and E2 included a morphological study to characterize the growth response of 8-day-old tomato seedlings grown in vitro under the tested conditions. The seedlings were at the same stage of development with fully expanded cotyledons at stage 10–100 [53] (Appendix A). After five days of hydroponic cultivation, some of the seedlings treated with NaCl appeared yellow and did not grow (Figure 1b). The mortality rate was 32% at 100 mM NaCl and 20% at 50 mM NaCl (Figure 1c). The control group did not experience any adverse effects, and the mortality rate was below 5% in the group treated with the AMM solution. Adding the AMM solution to the saline treatment decreased mortality rates to 16% in 100 mM NaCl and 4% in 50 mM NaCl (Figure 1c). The saline treatment significantly reduced the viability of the tomato seedlings (*p*-value = 0.000). The presence of AMM had a significant positive effect on viability under stress conditions at 100 mM NaCl (*p*-value = 0.0036). For experiments E3 and E4, we utilized 13-day-old seedlings that had the first pair of leaf primordia visible at stage 11–101 [53]. To quantitatively determine the impact of different treatments on seedling growth, we used new leaf formation under hydroponic conditions as a growth estimator (Figure 1d,e). A significant increase in the percentage of new leaf formation was observed after 5 days under hydroponic conditions in samples treated with AMM, in both 8-day-old and 13-day-old seedlings. Also, a significant increase in the leaf formation percentage was observed in AMM-treated plants at 100 mM NaCl compared with the same salt stress condition without treatment, while no significant effect of AMM was observed under the 50 mM NaCl condition (Figure 1d).

To evaluate the effect of the different treatments on root system growth, we measured several parameters, including primary root (PR) length, lateral root (LR) number, and adventitious root (AR) number, in 8-day-old seedlings grown for 5 days in hydroponic conditions (Figure 2a). The combined treatment of AMM solution with 50 or 100 mM NaCl significantly (*p*-value = 0.0087) reduced PR length, as well as with 50 mM salt stress compared to the other conditions (Figure 2b). A higher rate of LR formation was observed under the 50 or 100 NaCl salt stress conditions compared to the control conditions, while the AMM treatment did not significantly change the average LR number (Figure 2c). This higher rate of LR formation in both salt stress conditions also corresponded with an increased number of LRs produced compared with the control condition (Figure 2d). In the control conditions, we did not observe the appearance of ARs from the hypocotyl, while we found a positive and significant effect of either the AMM solution or salt stress (50 and 100 mM) on AR emergence (*p*-value = 0.0078). The percentage increase in AR capacity was statistically significant in both AMM solution and salinity conditions (Figure 2e). The AMM treatment promoted the formation of up to five ARs in comparison to the mock conditions, as well as in the 100 mM NaCl salt stress condition or the combination of AMM plus 100 mM NaCl. The number of ARs was highly variable between the samples and conditions, with the highest average value observed with 100 mM NaCl (average ARs = 1.0 ± 0.3) (Figure 2f), despite no individual showing more than six roots.

### 2.2. Gene Expression Analysis in Response to AMM and Moderate Salinity

To elucidate the molecular pathways that are modulated in response to AMM and moderate salinity (50 mM NaCl) in our experimental setup, a transcriptome analysis of polyadenylated RNA in leaves was performed (see Section 4). The results obtained included the expression data of 21,275 protein-coding genes and 908 non-coding RNAs (ncRNAs) (Appendix A). Principal component analyses indicated that most of the observed variability in gene expression data between samples was due to the treatment, validating our experimental design (Figure 3a). To further characterize the gene expression profiles, we performed a k-means analysis on the most variable genes (n = 6600), which yielded four distinct expression clusters (Appendix A). The analysis revealed that genes with an upregulated expression in response to moderate salinity were predominantly distributed across clusters 1 and 3, with low effect of the AMM treatment (Appendix A). Genes within cluster 2 were found upregulated in response to the AMM treatment and downregulated by moderate salinity, while those in cluster 4 were found downregulated in all three treatments under the mock conditions (Appendix A). Gene Ontology (GO) enrichment was performed in genes from these four clusters, and the results are indicated in Appendix A. We identified 3658 differential expressed genes (DEGs) in four different contrasts (Figure 3b and Appendix A), representing 17.2% of all expressed genes. The UpSet plots of DEGs in the different contrasts allowed us to identify either treatment-specific or shared genes between the treatments (Figure 3c).

Among the DEGs that were found upregulated in response to the AMM treatment (C1 contrast), we observed a significant enrichment of GO terms associated with a defense response (Figure 4a and Appendix A). Among these genes, we found *Solyc06g070990.3* and *Solyc03g095770.3* (Figure 4b and Appendix A), which encode two transcription factors of the WRKY family [54], *Solyc03g121900.1* encoding the putative apoplastic lipid transfer protein DEFECTIVE IN INDUCED RESISTANCE 1 (Figure 4b and Appendix A) involved in systemic acquired resistance in *Arabidopsis thaliana* [55], and *Solyc10g007600.3* encoding a glycolate oxidase, GOX3 (Figure 4b and Appendix A), that is known to modulate reactive oxygen species (ROS)-mediated signal transduction during non-host resistance [56]. It is hypothesized that these genes could regulate the systemic responses to root colonization by the AMF and PGPR present in the AMM solution. Conversely, downregulated genes in the leaves of the AMM-treated seedlings were associated with a response to oxidative stress, particularly linked to hydrogen peroxide metabolism (Figure 4a and Appendix A). It is notable that the expression levels of genes *SlPrx52* (*Solyc08g069040.3*) and *SlPrx102* (*Solyc12g005370.2*), which encode two peroxidases of the class III, were strongly downregulated by the AMM treatment, even in the presence of moderate salinity (Appendix A). These results are consistent with the observed establishment of an effective symbiosis between the AMM and the tomato roots, which can be detected transcriptionally in distant tissues, such as leaves, using some of these specifically deregulated genes as biomarkers.

In response to moderate salinity (C2 contrast, cluster 1), upregulated DEGs were significantly enriched for GO terms related to photosynthesis (light-dependent reactions) and response to abiotic stimulus (Figure 4c and Appendix A). We identified 82 expressed genes assigned to photosynthesis-related KEGG pathways, and the expression of 44 of these genes (53.7%) was deregulated upon moderate salinity (Appendix A). A significant increase in the expression of key genes in the different photosynthetic complexes was observed in response to the different conditions, which exhibit an additive effect in AMM treatment and salinity (Appendix A), Interestingly, among the DEGs that were downregulated by moderate salinity, genes related to defense response were also found (Figure 4b). Several of these salt-downregulated genes, including *Solyc06g070990.3* and *Solyc03g095770.3*, encode WRKY transcription factors that were found highly upregulated by the AMM treatment (Figure 4b and Appendix A), suggesting some antagonism between AMM treatment and moderate salinity. Two genes encoding two 9-cis-epoxycarotenoid dioxygenase (NCED) enzymes involved in abscisic acid (ABA) biosynthesis were identified—*NOTABILIS* (*NOT*; *SlNCED1*; *Solyc07g056570.1*) and *SlNCED2* (*Solyc08g016720.1*). These genes showed a significant decrease in expression in response to moderate salinity in the tomato leaves (Appendix A). In addition, *SlCCD8* (*Solyc08g066650.3*), a gene involved in strigolactone biosynthesis, also showed a significant decrease in expression under moderate salinity conditions in the tomato leaves (Appendix A).

In this regard, we have annotated 117 genes related to the plant hormone transduction pathway in KEGG (Appendix A). Of particular interest are *Solyc08g021820.3*, which encodes the IAA29 corepressor of auxin signaling [57], *Solyc10g076410.1*, which encodes the ABA receptor SlPYL10 [58], and *Solyc12g009220.2*, which encodes SlJAZ2 [59]. These genes exhibited a substantial increase in expression in response to AMM and were potently repressed by salt (Figure 4d and Appendix A).

Recent examples have demonstrated the effectiveness of AMF and/or PGPR inoculation of seeds/seedlings in enhancing salt stress tolerance in various crops [60,61,62]. To ascertain the combined effect of AMM treatment and moderate salinity in our experimental setup, the DEGs obtained in the C3 and C4 contrasts were analyzed in detail (Appendix A). It was determined that, among the DEGs that were found to be expressed at higher levels in both contrasts, there was a highly significant enrichment of GO terms related to de novo post-translational protein folding and ROS response (Appendix A).

In order to validate the results obtained in our RNA-Seq, an additional experiment was carried out. This experiment entailed reverse transcription, followed by quantitative PCR (RT-qPCR), utilizing RNA obtained from plant samples that had been frozen (see Section 4). A selection of five DEGs that exhibited differential expression were chosen, including the upregulation in AMM treatment (*IAA29*, *WRKY53*) or in AMM treatment and moderate salinity (*GLO2*), as well as *Prx52* and *Prx102*, which were downregulated under AMM treatment and moderate salinity conditions (Appendix A). The results of the RT-qPCR analysis demonstrate a congruent expression pattern with those obtained through RNA-Seq (Appendix A). In addition, a high degree of correlation is observed between both datasets (Appendix A).

Our transcriptomic analysis suggested that moderate salinity combined with AMM inoculation triggered the expression of several key genes related to the unfolded protein response, which is a cellular response that aims to restore protein homeostasis in the endoplasmic reticulum after prolonged stress [63]. It is noteworthy that this specific response does not appear to be highly activated by AMM treatment or moderate salinity alone. Conversely, a subset of downregulated DEGs in the combination of AMM treatment and moderate salinity were associated with water transport, including some aquaporin-encoding genes, such as *SlTIP2;3* (*Solyc06g060760.3*), *SlTIP1;2* (*Solyc06g075650.3*), and *SlPIP2;4* (*Solyc06g011350.3*) (Appendix A). These findings collectively imply that root inoculation with AMM solution could regulate water loss in a non-cell autonomous manner under moderate salinity conditions, as previously described with tomato plants treated with *Azotobacter chroococcum* [64].

### 2.3. Regulatory Responses to AMM and Moderate Salinity

In a previous study, we annotated 1940 tomato genes encoding transcription factors (TFs) that were classified into 48 families [65]. In the present study, we identified differential regulations of 254 of the 1326 TF-encoding genes expressed in our experimental setup (Appendix A). Hierarchical clustering and K-means analysis of these DEGs revealed significant disparities in their expression patterns, which were categorized into three clusters (Figure 5a). Cluster 1 comprised 96 transcription factor (TF) genes that were repressed in all conditions tested relative to control conditions. Cluster 2 included 77 TF-encoding genes that increased in expression in samples treated with AMM and decreased in the rest, including AMM and moderate salinity. Cluster 3 had 81 TF-encoding genes that increased under salinity conditions, both with and without AMM (Figure 5a). These findings confirm that AMM treatment does not reverse the transcriptional effect of moderate salinity. The enrichment of these DEGs has been estimated according to the TF family they encode and as a function of the contrast performed (Figure 5b). It is evident that certain TF-encoding families, including ERF, C2H2, MYB, NAC, and bHLH, exhibit members in numerous contrasts with comparable frequencies. Conversely, families such as Dof, WRKY, and bZIP exhibit a concentration of deregulated members in specific contrasts (Figure 5b). It is important to examine the case of TF-encoding DEGs belonging to the WRKY family, where a considerable proportion of these genes exhibit a substantial increase in expression in response to AMM treatment compared to the control conditions. However, the levels of these genes are found to be repressed by saline treatment, irrespective of the AMM treatment (Figure 5c). In contrast, the TF-encoding genes of the bZIP family show a significant increase in expression in response to salinity, except for bZIP62 which shows a pattern like WRKY (Figure 5c).

In the present study, we identified 908 non-coding RNA sequences that showed consistent expression according to our experimental design (Appendix A). We found that 78 ncRNAs (8.6% of expressed ncRNAs) exhibited differential expression in some of the treatments tested (Appendix A), and hierarchical clustering revealed the following dynamic expression patterns: (cluster i) 19 ncRNAs appeared to be downregulated in all treatments (AMM; 50 mM NaCl; AMM and 50 mM NaCl) compared to the mock conditions; (cluster ii) 6 ncRNAs showed downregulation of their expression only under moderate salinity conditions; (cluster iii) 21 ncRNAs exhibited an increase in expression in response to the AMM treatment but not in moderate salinity or AMM plus salinity; and (cluster iv) 24 ncRNAs showed higher expression levels in moderate salinity and AMM plus salinity conditions. A further eight ncRNAs exhibited an atypical expression pattern (Appendix A). Of the 78 degraded ncRNAs, 37 were classified as antisense, 31 as intergenic and the remaining 10 corresponded to exonic or intronic ncRNA. No significant enrichment in ncRNA type compared to the total expressed ncRNAs was observed. In addition, we did not find any association between specific ncRNA, and the protein coding genes located in the same genetic locus.

Despite the absence of evidence supporting the functionality of these ncRNAs, the stringent regulation of their expression under the conditions examined suggests a potential for involvement in the regulation of gene expression in response to stress. Further research is warranted to investigate this possibility.

## 3. Discussion

Salt stress represents one of the most pernicious abiotic stresses, exerting deleterious effects on plant growth and productivity. These conditions trigger a series of physiological responses, including stomatal closure and the generation of ROS. Other notable changes encompass metabolic imbalance and photosynthesis limitation, collectively impacting plant growth and development [66]. Salinity is known to limit plant growth and productivity; however, the effect of salinity on tomato growth can be highly variable. In a recent study [67], it was observed that different concentrations of NaCl had different effects on the *S. lycopersicum* cultivar ’Moneymaker’ grown in vitro. Specifically, low-to-moderate salt concentrations (25–50 mM) enhanced lateral root development and modified other morphological characteristics, including shoot fresh weight. In contrast, higher salt concentrations (>100 mM) in the culture medium had been shown to exert a negative effect on seedling growth [67].

A hydroponic system was utilized to cultivate and assess the impacts of diverse compounds and stresses on juvenile tomato seedlings (Appendix A). This system facilitated the precise regulation of experimental parameters and provided sufficient biological material for omics studies. The present study investigated the effects of moderate salinity in the culture medium on various growth parameters, thereby enabling its application in the evaluation of other stresses, including nutrient deficiency, temperature fluctuations, or simultaneous exposure to multiple stressors. Additionally, the impact of incorporating biostimulants to foster effective symbiosis with the root system under these conditions was examined, with the objective of leveraging this experimental system for quantitative assessment in the identification of novel biostimulant compounds. The methodology established in this study, with appropriate modifications, has the potential to serve as the basis for the evaluation of other plant species, such as lettuce or pepper.

Observations were made regarding the impact of moderate salinity on the seedlings of the tomato cultivar ’Rosa de Altea’, a traditional variety from La Marina Baixa (Alicante, Spain) with a certain tolerance to salinity due to its cultivation near the Mediterranean coast. Moderate salinity has been shown to exert a deleterious effect on the growth of the primary root and to induce the development of lateral and adventitious roots in juvenile tomato seedlings in our experimental conditions (Figure 1 and Figure 2). The inoculation of these seedlings with the AMM biostimulant, containing a proprietary mixture of mycorrhizal microorganisms and rhizobacteria, facilitates the establishment of a robust root symbiosis. This results in an augmentation of the root system area, characterized by an increase in the number and length of lateral and adventitious roots (Figure 2). Moreover, the AMM treatment has been observed to mitigate the deleterious effects of moderate salinity conditions on seedling viability (Figure 1), a finding that is consistent with the results of other research groups who have demonstrated that inoculation with AMF or PGPR enhances the response of tomato plants to salinity [45,48,68,69].

To validate the use of our experimental approach for transcriptomic analysis, we performed an assay with samples in control conditions, treated with the biostimulant (AMM), in moderate salinity conditions (50 mM NaCl), and in the combination of both factors (AMM and 50 mM NaCl). The analysis revealed that approximately 74.9% of the observed variability in the expression of protein-coding genes and non-coding RNAs could be attributed to the treatment effect, with a mere 5.6% being attributable to experimental replicates (Figure 3a,d and Appendix A).

In the present study, we identified differential expression in 3658 genes in the leaves of tomato seedlings grown in vitro under three different conditions (AMM, NaCl, and AMM-NaCl), as regards the control conditions (Figure 3b,c and Appendix A). These genes were found to be enriched in processes related to photosynthesis, response to stress (both abiotic and biotic), hormone regulation, and response to ROS, among others (Figure 4 and Appendix A). The moderate salinity treatment elicited the most substantial expression changes in the transcriptome, exerting a highly significant effect on photosynthesis-related and abiotic stress response genes (Appendix A). The saline treatment increased the expression of these genes, both in the absence and presence of the biostimulant, which agrees with the mild improvement observed in on salinity-treated seedlings treated with AMM in our study. The upregulation of photosynthesis genes in salt-treated tomato leaves was counterintuitive, given the reduced growth observed due to salt stress (Figure 1 and Figure 2; [70]), and the known effect of salt stress on lowering the photosynthetic induction rate and limiting carbon fixation [71]. We hypothesize that the upregulation of these photosynthesis-related genes by moderate salinity may imply a regulatory mechanism to compensate for the damage to chloroplasts caused by salt stress. This hypothesis is further supported by been recent findings in *Robinia pseudoacacia* seedlings [72]. Additionally, two genes required for ABA biosynthesis—(*SlNCED1*) and *SlNCED2*—were found to be downregulated by moderate salinity in young tomato leaves. A recent study has demonstrated that *SlNCED1*, *SlNCED2*, and *SlNCED6* are strongly induced under severe salt stress (94 mM NaCl) in tomato cultivar ’Ailsa Craig’, while this upregulation was not observed in young leaves at moderate salt levels (47 mM NaCl) [73]. Considering the observation that the tomato variety used in our study (’Rosa de Altea’) displays tolerance to moderate salinity, the low expression levels of these *NCED* genes may be indicative of an adaptive response. However, further studies are needed to determine the endogenous levels of ABA in young tomato leaves in our experimental system to test this hypothesis.

The present study offers evidence that increased salinity levels could render plants more susceptible to pathogen attack or exert a negative influence on mycorrhization. This phenomenon is characterized by a decrease in the expression by moderate salinity of defense genes and the *SlCCD8*, which is involved in the biosynthesis of the strigolactone hormones required for effective mycorrhization [74]. The latter finding could offer a potential explanation for the increased incidence of adventitious roots observed in tomato plants cultivated under saline conditions [75]. The interplay between jasmonic acid (JA) and auxin signaling has been extensively studied in various developmental processes and in response to stress [76]. The ability to JA-induced leaf senescence to be antagonized by auxin via the transcription factor WRKY57, which has been shown to interact with JASMONATE ZIM-DOMAIN4/8 (JAZ4/8) and the AUX/IAA protein IAA29, repressors of the JA and auxin signaling pathways, respectively [57], is of particular interest. In addition, WRKY53 has been shown to negatively regulate the JA biosynthesis pathway, thereby contributing to plant defense mechanisms in Arabidopsis [77]. In the present study, we identified that multiple WRKY genes were found to be upregulated by the biostimulant treatment, with *Solyc08g008280.3*, a putative orthologue encoding WRKY53, being among the most expressed (Appendix A). Furthermore, *Solyc08g021820.3*, putatively encoding IAA29 (Appendix A), and *Solyc12g009220.2* which encodes SlJAZ2, exhibited increased expression in biostimulant-treated samples and were found to be inhibited by the saline treatment. These findings imply the presence of a JA and auxin cross-regulation module in tomato involving WRK53, IAA29, and JAZ2, which could be adversely affected by salinity. This pathway will be the subject of further research.

## 4. Materials and Methods

### 4.1. Plant Materials

The seeds used in this study, *Solanum lycopersicum* L. var. ‘Rosa de Altea’, were kindly provided by Dr. Santiago García Martínez (Universidad Miguel Hernández de Elche, Spain). The biostimulant solution used, Atlanticell Micomix (AMM), was obtained from Atlántica Agrícola S.A. (Villena, Spain). The AMM solution is a proprietary biological stimulant composed of mycorrhizal microorganisms (*Rhizoglomus irregulare, Funneliformis mosseae* and *Funneliformis caledonium*; 12,500 propagules/g) and rhizobacteria (*Bacillus* spp., equivalent to a concentration of 1 × 10^10^ UFC/g).

### 4.2. Culture of Tomato Seedlings

The seeds underwent surface sterilization using chlorine gas (100 mL NaOCl 10% and 4 mL HCl 37%) for 3.5 h in an airtight container. Next, the seeds were transferred to multi-well plates with 25 seeds per well and soaked in sterile water for 24 h in darkness inside a growth chamber at 25.4 ± 1.6 °C. The following day, the germinated seeds were transferred to square Petri dishes (120 mm × 120 mm × 10 mm). Each Petri dish contained 75 mL of 2.15 g/L of Murashige & Skoog (MS) salt medium, 2.5 g/L of gelling agent Gelrite (Duchefa Biochemie, Haarlem, The Netherlands), 0.5 g/L 2-(N-morpholino) ethanesulfonic acid (Duchefa Biochemie), and 2 mL/L Gamborg B5 vitamin solution (Duchefa Biochemie) at pH 5.8. The seedlings grown in vitro were incubated in a growth chamber under controlled conditions, with a photoperiod of 16 h light (day) and 8 h darkness (night) until the end of the experiment provided by white and red LED panels. The average temperature during the day was 25.4 ± 1.6 °C and during the night was 22.8 ± 1.3 °C. The average relative humidity during the day was 49.5 ± 9.1% and during the night was 64.6 ± 12.2%. A graphic visualization of the procedure is shown in Appendix A.

Four different experiments were performed, named E1 to E4. Eight (E1, E2) or thirteen (E3, E4) days after germination, the seedlings from the same developmental stage were transferred to perforated plastic floats positioned above 1 L of liquid culture medium in square plastic containers (165 × 165 × 135 mm), with enforced aeration using an aquarium air pump. The liquid medium was prepared as indicated above, without the addition of the gelling agent Gelrite (Duchefa Biochemie). The salt treatment consisted of 100 mM NaCl (E1 experiment) or 50 mM NaCl (E2 to E4 experiments). The AMM solution, containing a mixture of AMF and rhizobacteria (see above), was added to the liquid culture medium at a concentration of 1 g/L. The following four different conditions were tested: mock, salinity, AMM, and salinity plus AMM. The number of plant samples ranged from 17 to 25 for each condition. The nutrient solution was renewed every two days. All the four experiments were conducted between April and May 2021.

### 4.3. Sample Collection and RNA Extraction

For transcriptomic analysis, three experimental replicates were taken for each condition (mock, salinity, AMM, salinity plus AMM) from E3 and E4 experiments. Each replicate consisted of all aerial tissues, including hypocotyl, cotyledons, and leaves, from six different plants. The samples were collected five days after growing in hydroponic culture, immediately frozen in liquid nitrogen, and stored at −80 °C. Total RNA was extracted from approximately 100 mg of powdered tissue using TRIzol reagent (Invitrogen, Carlsbad, CA, USA), following the manufacturer’s protocol in duplicate (RNA-Seq and RT-qPCR). The RNA was then stored at −80 °C. The integrity of the RNA was assessed using a 2100 Bioanalyzer (Agilent Technologies, Santa Clara, CA, USA). Directional RNA sequencing was performed in (BGI-Tech, Shenzhen, China) using DNB-Seq technology. Sequencing was carried out in paired-end mode with 150 sequencing cycles. The reads were processed using SOAPNuke to obtain the clean reads [78]. Three libraries were sequenced from E4 plants in each condition. The raw data will be provided upon request to the corresponding author.

### 4.4. RNA-Seq Analysis

The bioinformatics workflow utilized in this study was based on the previously described protocol by [79]. In brief, cleaned RNA-Seq reads were mapped to the *S. lycopersicum* genome build SL4.0 [80] using STAR 2.7 [81]. This approach resulted in 92% of average reads from different libraries mapped to the SL4 genome (Appendix A). Gene count was performed using featureCounts from the Subread package [82] with the ITAG4 gene models from SolGenomics [83]. Raw counts were transformed to CPM for normalization (Appendix A), and genes with less than 0.5 CPM in at least three libraries were excluded. A correlation matrix was used to analyze the relationship between the samples (Appendix A). DESeq2 integrated into the Differential Expression and Pathway analysis (iDEP 2.0) web application was used for read count normalization and differential gene expression analysis [84]. All DEGs were obtained using an FDR threshold of >0.05 and a fold change threshold of >|2|. GO enrichment analysis was performed using the iDEP 2.0 [85]. Putative tomato orthologs from *A. thaliana* were identified using Proteinortho [86]. For the identification of expressed and deregulated transcription factors, we used those annotated in [65]. The fold enrichment of the different TF families was obtained by applying the following formula: (number of DEGs up or down of a TF family/total number of TF DEGs up or down)/(number of expressed genes of a TF family/total TF genes expressed). Plant hormone signal transduction was obtained from the KEGG database (January 2025). For heatmap and hierarchical clustering analyses, we used Morpheus web tool [87]. The ncRNAs annotation was primarily retrieved from the SolGenomics database. The JustRNA database was also consulted for ncRNA annotation (January 2025) [88].

### 4.5. Reverse Transcription-Quantitative PCR (RT-qPCR)

RNA samples were subjected to a retrotranscription using RevertAid (ThermoFisher Scientific, Waltham, MA, USA), in accordance with the manufacturer’s instructions. The cDNA obtained was adjusted to 20 ng/µL, and 1.5 µL was used as a template for quantitative PCR, using the PowerTrack SYBR Green kit in a QuantStudio 3 Real-Time PCR System (ThermoFisher Scientific, Waltham, MA, USA). To calculate the PCR efficiency of each primer, a linear regression equation was calculated for each target using serial dilutions of a pool of cDNA. Subsequently, a melting curve analysis was performed for each reaction to verify the synthesis of a single PCR product. The analyses described above, in addition to the quality control analysis, the determination of the baseline, the interplate adjustment, and the calculation of the relative expression values (RQ) using *GADPH* as the endogenous reference gene, were performed with the Standard Curve and Relative Quantification apps in the Connect Data Analysis platform (ThermoFisher Scientific, Waltham, MA, USA). The primers used for RT-qPCR are indicated in Appendix A.

### 4.6. Microscopy Observation

Basal root fragments (approximately 5 mm) were obtained using a razor blade from hydroponically grown seedlings either in mock condition or in the AMS treatment from E1 and E2 experiments (n = 5 each). The root fragments were then placed separately in multi-well plates and stained with trypan blue solution (ThermoFisher Scientific, Waltham, MA, USA) as follows: 2 mL of 10% KOH was added to each well, and the plate was incubated at 85 °C for 10 min. The KOH was then removed, and the roots were rinsed twice with 2 mL of distilled water. The distilled water was removed, and 2 mL of 2% HCl were added to each well. The plate was incubated at room temperature (RT) for 15 min. Next, the HCl solution was removed, and 2 mL of 0.05% trypan blue were added to each well and incubated at RT for 30 min. The trypan blue solution was then removed, and the roots were rinsed four times with 2 mL of distilled water. The samples were observed using a Motic BA210 brightfield microscope (MoticEurope, SLU Barcelona, Spain) and imaged with an integrated Moticam 580INT documentation station (Motic Spain).

### 4.7. Trait Measurement

We quantified the number of viable seedlings and those producing true leaves at 13 days after germination (5 days after transfer to hydroponic conditions). Twenty days after germination (12 days after transfer to hydroponic conditions), all seedlings from E1 and E2 experiments were carefully placed on a large square Petri dish (245 × 245 × 20 mm) with 150 mL of tap water. Pictures of these seedlings were taken using a photographic bench with a Sony Cyber-shot DSC-H3 8.1 Megapixel digital camera (Sony Corporation, Tokyo, Japan) at a resolution of 3264 × 2448 pixels and saved in JPEG format. Several parameters of the root system of these plants were quantified from these pictures as described elsewhere [89], including primary root length, lateral root number, and the presence of adventitious roots in the hypocotyl.

### 4.8. Statistical Analyses

Statistical analyses were conducted with StatGraphics Centurion XV software v.16.1.03 (StatPoint Technologies, McKinney, TX, USA). Descriptors such as mean, standard error (SE), maximum, minimum, and correlation values were estimated. Data outliers were identified based on aberrant standard deviation values and were excluded from downstream analyses. To compare the data for a given variable, we conducted multiple testing analyses using either the ANOVA F-test or the Fisher’s least significant difference method. We defined significant differences as a 1% level of significance (*p*-value < 0.01), unless otherwise specified. To establish correlations between different parameters, we conducted multiple correlation tests for the selected parameters.

## 5. Conclusions

The present study explores the molecular and phenotypic changes in tomato plants in response to salt stress adaptation through the AMM biological stimulant compound. The in vitro hydroponic system enabled observation of the treatment effects on plant viability, growth rates, and root system changes. The AMM compound enhanced tomato plant growth under both control and salt stress conditions. A detailed analysis of the root system revealed a decrease in primary root length and an increase in lateral root length under salt-stressed conditions. Notably, the AMM-treated plants exhibited no significant variation, although seedling viability was enhanced in comparison to those under salt stress conditions. Furthermore, AMM treatment significantly enhanced AR development, demonstrating a comparable response to salt stress conditions.

The transcriptomic studies conducted in this research facilitated the identification of over 3000 differentially expressed genes. Our findings revealed that several WRKY genes were induced by AMM treatment, including a putative ortholog encoding WRKY53, which was particularly noteworthy for its high level of expression. Furthermore, several other genes, including potential orthologs of *IAA29* and *SlJAZ2*, exhibited increased expression in AMM-treated samples but were repressed by salt treatment. These observations suggest the presence of a cross-regulatory module between JA and auxin in tomato, involving these genes, which could be adversely affected by salinity. Moreover, treatment with the AMM compound led to an increase in the expression of genes such as *GOX3* or *DIR1*, which are associated with the plant’s defense response to adverse conditions through the ROS detoxification pathway or the salicylic acid-mediated immune response, respectively. The activation of these genes suggests a priming effect of the AMM biostimulant in tomato that deserves further studies.

## Figures and Tables

**Figure 1 plants-14-01198-f001:**
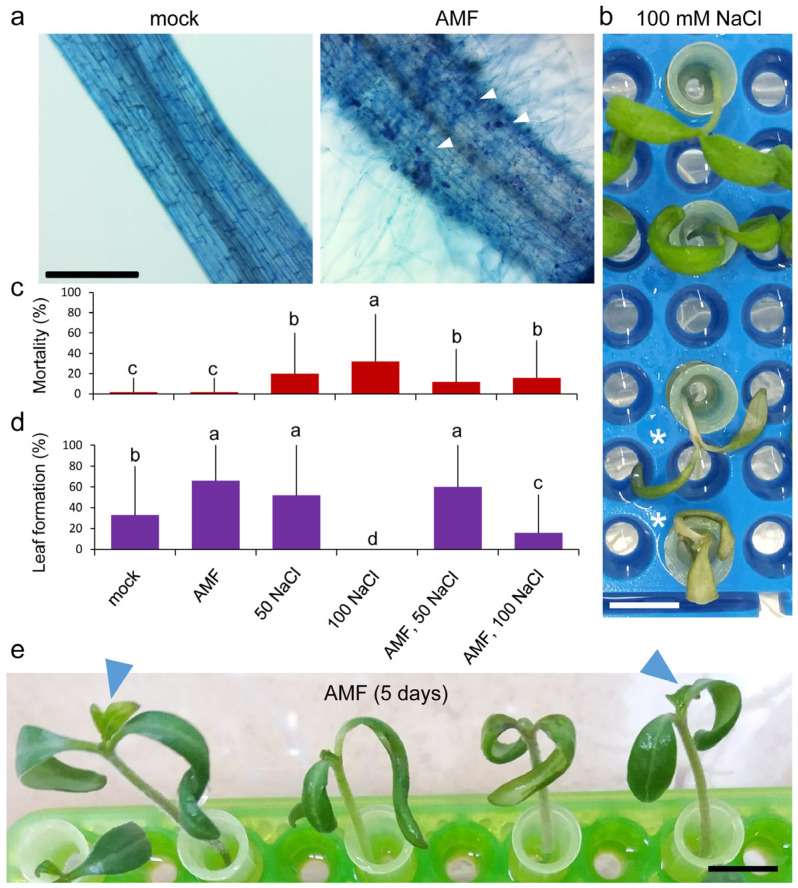
Colonization and shoot morphology of Atlanticell Micomix (AMM)-treated plants under salt stress. (**a**) Representative images of trypan blue-stained roots from *S. lycopersicum* var. ‘Rosa de Altea’ seedlings grown for 5 days in hydroponics; white arrowheads indicate some intracellular arbuscules in the cortical layer. (**b**) Detailed view of the shoot morphology of salt-treated seedlings; asterisks indicate yellow plants that did not develop further. (**c**) Percentage of seedling mortality in the different conditions assayed. (**d**) Percentage of new leaf formation in the different conditions assayed. Letters in (**c**,**d**) indicate statistically significant differences (*p*-value < 0.05; Chi-Squared test). (**e**) Representative images of tomato seedlings at hydroponic culture. Blue arrowheads indicate the new leaf formation. Scale bars: (**a**) 0.5 mm, and (**b**,**e**) 5 mm.

**Figure 2 plants-14-01198-f002:**
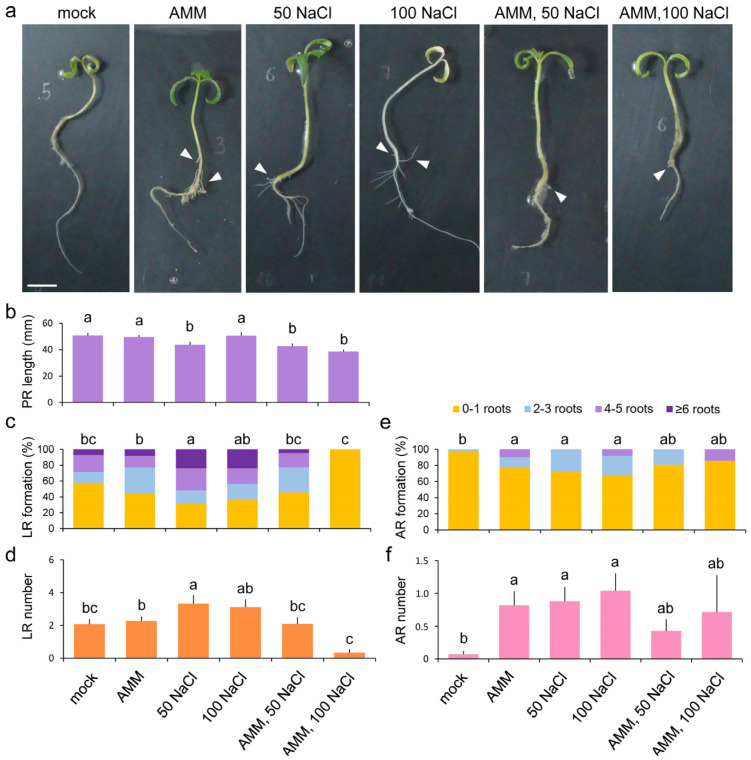
Root morphology of Atlanticell Micomix (AMM)-treated plants under salt stress. (**a**) Representative images of *S. lycopersicum* var. ‘Rosa de Altea’ seedlings grown for 5 days in hydroponics; white arrowheads indicate some adventitious roots (AR). (**b**) Average primary root (PR) length in the different conditions assayed. (**c**,**e**) Percentage of lateral roots (LR) (**c**) and AR (**e**) formation in the different conditions assayed. (**d**,**f**) Distribution of LR (**d**) and AR number (**f**) in the different conditions assayed. The lines in bar graphs indicate standard deviation. Letters in (**b**–**f**) indicate statistically significant differences (*p*-value < 0.05; Fisher’s least significant difference). Scale bar: 10 mm.

**Figure 3 plants-14-01198-f003:**
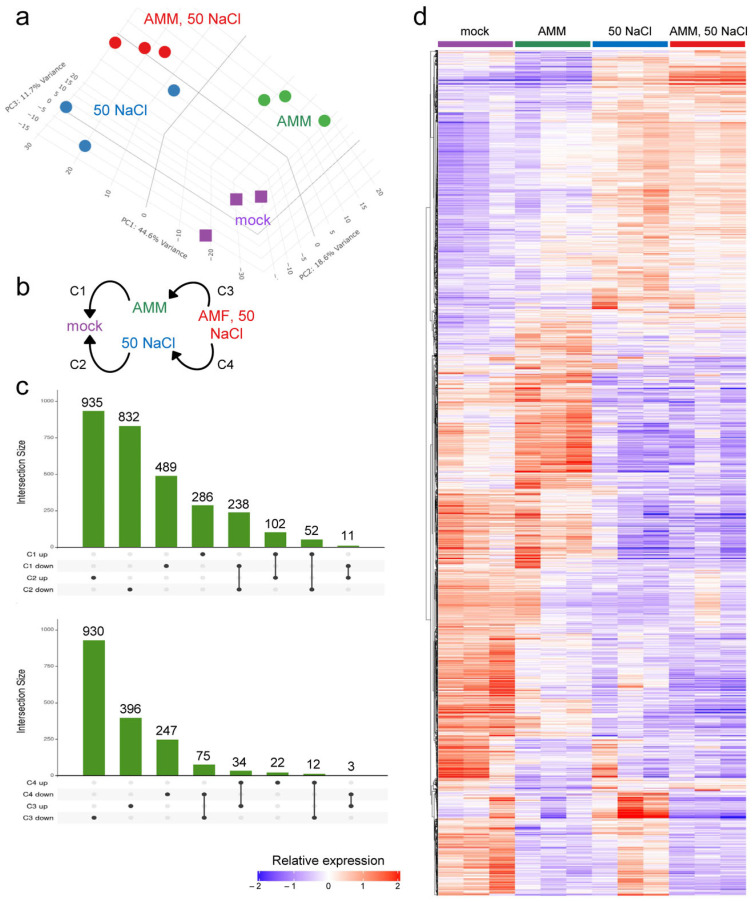
Expression analysis of RNA-Seq data. (**a**) Principal component analysis of the RNA-Seq results from protein coding genes. (**b**) Diagram of the four comparisons (C1 to C4) carried out for the analysis of DEGs. (**c**) UpSet plot of the number of DEGs from indicated contrasts. (**d**) K-means clustering of DEGs. Predicted clusters are grouped according to their expression profile. Expression values are relative and adjusted to −2 and +2.

**Figure 4 plants-14-01198-f004:**
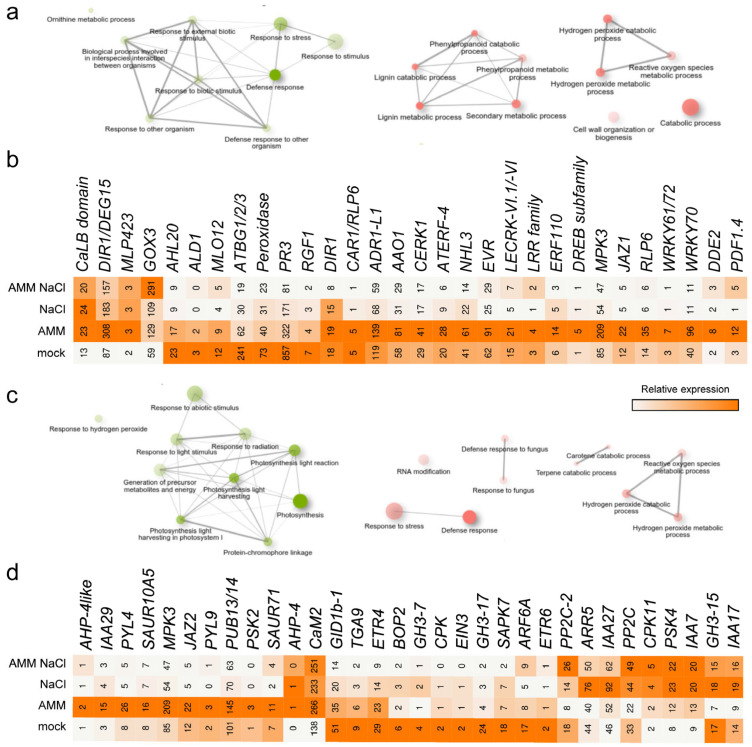
Functional annotation of differentially expressed genes (DEGs). (**a**,**c**) The interaction networks of the different termini in Gene Ontology Biological Process (GO BP) significantly enriched (False discovery rate = 0.05) in the upregulated (green) and downregulated (red) genes in C1 (**a**) and C2 (**c**) contrasts. (**b**,**d**) DEGs involved in defense response (**b**) and phytohormone signaling (**d**). Grey/orange in (**b**,**d**) indicate normalized counts per million (CPM) values in each row, where orange indicates the highest abundance of transcripts. Gene annotations are found in Appendix A.

**Figure 5 plants-14-01198-f005:**
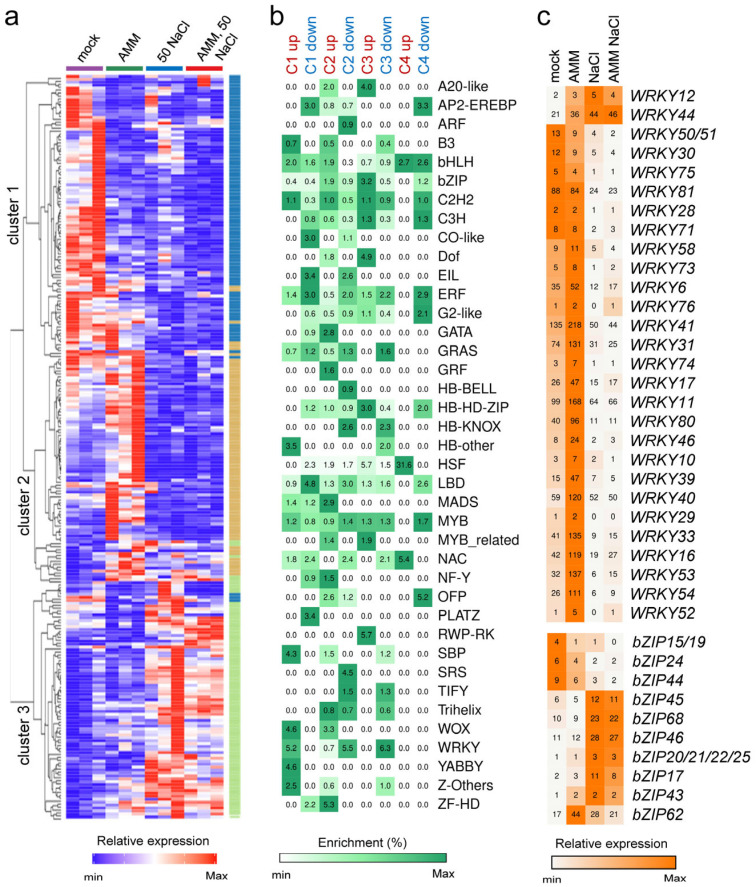
Expression analysis of transcription factor genes. (**a**) K-means clustering of differentially expressed genes (DEGs). Predicted clusters are grouped according to their expression profile. Expression values are relative and adjusted to −2 and +2. (**b**) Fold enrichment of the different transcription factors (TF) families expressed; the intensity of the green color indicates the strength of the fold enrichment (**c**) TF-encoding DEGs belonging to the WRKY and bZIP families. Grey/orange indicate normalized counts per million (CPM) values in each row, where orange indicates the highest abundance of transcripts.

## Data Availability

All data generated or analyzed during this study are provided in this published article and its Appendix A or it will be provided at a reasonable request.

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
