# Peer review of "Transcriptional Profiling to Assess the Effects of Biological Stimulant Atlanticell Micomix on Tomato Seedlings Under Salt Stress"

_plants, 2025, doi:10.3390/plants14081198_

Round 1

Reviewer 1 Report

Comments and Suggestions for Authors

The utilization of plant biostimulants as alternatives to synthetic compounds to mitigate the inhibitory effects of abiotic stresses on plant growth and development, thereby enhancing the sustainability of agricultural production. This manuscript investigates, through transcriptomic analysis, the impact of Atlanticell Micomix (AMM, a proprietary biological stimulant) on hydroponically-grown tomato seedlings under salt stress. The findings provide preliminary insights into the mechanisms by which AMM regulates tomato seedling growth under salt stress conditions. The manuscript is well-written, with comprehensive research, accurate result analysis, thorough discussion, and standardized figures and tables. The primary deficiency lies in the absence of validation experiments for key differentially expressed genes (DEGs). The following comments are provided:

  1. The title should explicitly specify AMM.
  2. Line 16: The manuscript only presents results related to salt treatment.
  3. The abstract should include specific research findings.
  4. Standard deviations should be added to the data in Figures 1c and 1d.
  5. Figures 1b and 1e should use the same scale.
  6. The results in Figure 2b are inconsistent with those shown in Figure 2a.
  7. Validation experiments for specific DEGs, such as quantitative real-time PCR, are essential.
  8. The retrieval dates for relevant databases need to be indicated, as these databases are continuously updated.
  9. Abbreviations in Figures 4 and 5 require annotation.
  10. Table S4, as referenced in the manuscript, is missing from the supplementary materials.

Author Response

See atached PDF file the for point-by-point response letter.

Reviewer 2 Report

Comments and Suggestions for Authors

Review for „Transcriptional Profiling to Assess the Effects of a Biological Stimulant in Tomato Seedlings under Salt Stress”

The authors examined the effect of arbuscular mycorrhizal fungi and rhizobacteria on tomato root characteristics, leaf formation, and gene expression under salt stress conditions.

The authors concluded that arbuscular mycorrhizal fungi and rhizobacteria (AMM) inoculation increased tomato growth under salinity conditions. In addition, AMM treatment also increased seedling viability.

Importance of the study: Transcriptional profiling serves as an invaluable method for assessing the molecular impact of biological stimulants under salt stress conditions. This approach not only advances our understanding of plant stress responses but also aids in the development of innovative and sustainable solutions to mitigate the effects of soil salinization on crop productivity.

Weakness of the study: The manuscript does not contain the clear goal and hypothesis of the study.

Please present more precise data and do not only mention the „differential effect” in the Abstract.

Please use italics for in vivo.

Please arrange the keywords in alphabetical order.

Please do not start a sentence with an abbreviation. Please check the whole manuscript for this issue.

Lines 103-105. The last paragraph of the Introduction section must contain the clear goal and hypothesis of the study. Please add it.

Lines 120, 129, 205, 207, 252, 255, 264, 271, and 274: The title of the section is Results.

Figure 1: What does AMM refer to? Each figure and table must be understandable by itself without reading the whole manuscript. Please explain every presented abbreviation in the tables and figures.

Figure 2: Please explain the meaning of AMM, AR, PR, and LR.

Please focus only on research data and use citations in the Discussion section. In the Discussion section, the authors can compare their results with previous investigations/experiments.

Please refer to the presented data in the Results section in the Discussion section using Figure 1, Table 1, etc.

Line 450: Was the nutrient solution renewed or changed? If it was renewed, how did the authors ensure all the pots contained the same amount of the nutrients?

Some English improvement is needed. Please check the whole manuscript carefully and correct typo errors. 

Comments on the Quality of English Language

Some English improvement is needed. The authors should check the whole manuscript carefully and correct any typographical errors. 

Author Response

(The authors gave the same response as above.)

Round 2

Reviewer 2 Report

Comments and Suggestions for Authors

Thank you for the corrections.